# Combined Neutrophil-to-Lymphocyte and Platelet-Volume-to-Platelet Ratio (NLR and PVPR Score) Represents a Novel Prognostic Factor in Advanced Gastric Cancer Patients

**DOI:** 10.3390/jcm10173902

**Published:** 2021-08-30

**Authors:** Kamil Konopka, Agnieszka Micek, Sebastian Ochenduszko, Joanna Streb, Paweł Potocki, Łukasz Kwinta, Piotr J. Wysocki

**Affiliations:** 1Department of Oncology, Jagiellonian University Medical College, 31-007 Cracow, Poland; joanna.streb@uj.edu.pl (J.S.); pawel.potocki@uj.edu.pl (P.P.); lukasz.kwinta@uj.edu.pl (Ł.K.); piotr.wysocki@uj.edu.pl (P.J.W.); 2Department of Nursing Management and Epidemiology Nursing, Jagiellonian University Medical College, 31-007 Cracow, Poland; agnieszka.micek@uj.edu.pl; 3Department of Oncology, Dr. Peset University Hospital, 46017 Valencia, Spain; sebaochenduszko@gmail.com

**Keywords:** gastric cancer, NLR, PVPR, chemotherapy, inflammation, survival, prognosis

## Abstract

Background: Chemotherapy is a cornerstone of treatment in advanced gastric cancer (GC) with a proven impact on overall survival, however, reliable predictive markers are missing. The role of various inflammatory markers has been tested in gastric cancer patients, but there is still no general consensus on their true clinical applicability. High neutrophil-to-lymphocyte (NLR) and low (medium)-platelets-volume-to-platelet ratio (PVPR) are known markers of unspecific immune system activation, correlating significantly with outcomes in advanced GC patients. Methods: Metastatic GC patients (N:155) treated with chemotherapy +/− trastuzumab were enrolled in this retrospective study. Pre-treatment NLR and PVPR, as well as other inflammatory markers were measured in peripheral blood. Univariate Cox regression was conducted to find markers with a significant impact on overall survival (OS) and progression-free survival (PFS). Spearman correlation and Cohen’s kappa was used to analyze multicollinearity. Multiple multivariable Cox regression models were built to study the combined impact of NLR and PVPR, as well as other known prognostic factors on OS. Results: Elevated NLR was significantly associated with increased risk of death (HR = 1.95; 95% CI: 1.17–3.24), and lower PVPR was significantly associated with improved outcomes (HR = 0.53; 95% CI: 0.32–0.90). A novel inflammatory marker, based on a combination of NLR and PVPR, allows for the classification of GC patients into three prognostic groups, characterized by median OS of 8.4 months (95% CI 5.8–11.1), 10.5 months (95% CI 8.8–12.1), and 15.9 months (95% CI 13.5–18.3). Conclusion: The NLR and PVPR score (elevated NLR and decreased PVPR) is a marker of detrimental outcome of advanced GC patients treated with chemotherapy.

## 1. Introduction

Interactions between the immune system and cancer are attracting considerable interest because of growing evidence of underlying correlations that are sometimes too elusive to be appropriately described. However, many years of intensive evolution of the immuno-oncology field, which finally led to the introduction of checkpoint inhibitors, allowed us to predict, if not fully comprehend, the interplay between the immune system and malignancy in a way that is consistent and beneficial.

Despite significant progress in systemic treatment, metastatic gastric cancer (GC) remains incurable. In this palliative setting, systemic chemotherapy still represents the treatment of choice worldwide, with the potential to improve patients’ outcomes (overall survival and quality of life) [1].

Platinum-based chemotherapy is a cornerstone of the first-line treatment for stage IV gastric cancer [2]. Oxaliplatin demonstrates similar activity to cisplatin, but offers a much better safety profile [3]. Therefore, in combination with capecitabine [4], 5-fluorouracil [5] or S-1 [6] oxaliplatin is the most commonly used form of platinum derivatives for the treatment of GC. Most GC patients receive chemotherapy doublets [7], but less intensive chemotherapy regimens may be preferred in some circumstances due to e.g., suboptimal patient performance status [8]. In the past, tri-drug chemotherapy, such as a combination of EOX (epirubicin, oxaliplatin, and capecitabine) [9] or DCF (docetaxel, cisplatin, fluorouracil) [10], was considered the standard of care. Yet, more recent studies have not shown this approach to be more efficacious than other, less aggressive chemotherapy options [11]. For GC patients with HER2-overexpression, a combination of two-drug chemotherapy with trastuzumab is the treatment of choice [12]. 

Despite numerous efforts to incorporate immunotherapy into the first-line systemic treatment of GC [13], the optimal choice of chemotherapy regimen remains crucial in terms of patients’ outcomes. Even if doublet is an optimal regimen for most GC patients, some of them might benefit from more or less aggressive treatments. Therefore, to optimize the treatment strategy, optimal and robust predictive markers are of critical importance.

Systemic inflammation is a well-known phenomenon, although not fully understood, involved in carcinogenesis and progression of cancer, and the modulation of cancer cells’ immunogenicity and sensitivity to systemic treatment [14]. Therefore, quantifying systemic inflammation might improve our capability to predict the natural course of the disease and responses to systemic anti-cancer therapy.

The best apprehended and easy-to-measure marker of systemic inflammation is NLR (neutrophil-to-lymphocyte ratio), in which its prognostic potential was demonstrated in multiple malignancies [15,16,17]. High NLR exerts a significantly negative impact on the survival of both early [18] and advanced GC patients [19]. However, the level of its impact on prognosis is inconsistent through the various studies [20].

Many additional markers of systemic inflammation, such as PLR (platelet-to-lymphocyte ratio) [21], HPR (hemoglobin-to-platelet ratio) [22], PDW (platelet distribution width) [23], MPV/PLT (mean platelet volume to platelet ratio) [24], and RPR (red cell distribution width (RDW) to platelets ratio) [25] were described so far, but their true applicability in clinical practice is not well established. 

The PLR, PDW, and MPV/PLT are inflammatory markers reflecting the impact of platelet activation on the course of malignancies. In the majority of studies, high numbers of platelets correlated with poor prognosis [26]. This phenomenon can be caused both by secretion of proangiogenic, tumor-promoting factors but also by an increased risk of life-threatening thromboembolic events [27].

On the other hand, data on the prognostic and predictive potential of HPR and RPR factors incorporating red blood cell status are still scarce [28].

It is also to be noted that the best cut-off values for all of the above-mentioned inflammatory parameters are still unknown, and proposed values differ between studies [29].

The current study analyzed various systemic inflammatory parameters in 155 advanced GC patients treated with first-line therapy (plain chemotherapy without targeted agents). Our exploratory study aimed to define the optimal inflammatory marker with the highest predictive and prognostic potential.

## 2. Patients and Methods

Data on 155 GC patients treated in the Department of Clinical Oncology, University Hospital in Cracow, were retrospectively collected between September 2013 and December 2019. Data were censored on 31 December 2020. Patients were eligible for analysis if at least one cycle of palliative intravenous chemotherapy had been administered. Patients on steroid treatment, defined as a prednisone dose higher than 10 mg per day or equivalent, were excluded. The choice of chemotherapy regimen was left to the discretion of the leading physician; however, the decision was based on the patient’s general condition, HER2 expression, and current therapeutic guidelines at a given time. The allocation of patients to chemotherapy could have been biased by the evolution of guidelines, which occurred over time, promoting the switch from 3-drug to 2-drug regimens in asymptomatic or mildly symptomatic patients or patients with suboptimal performance status. The patient’s treating physician determined the moment of disease progression for each patient, based on clinical or radiological symptoms. The exact date of death was obtained from public databases. Due to the lack of the date of death for some participants, the date of the last visit was used in case of unequivocal progression and health deterioration. Peripheral blood inflammatory markers were measured at the baseline (up to one week before initiation of the first-line chemotherapy). The NLR was calculated by division of absolute neutrophile and lymphocyte counts. The RDW/PLT was defined as the ratio of RDW-SD (red cell distribution-standard deviation) to platelet count. The HPR was calculated by division of Hgb (hemoglobin) and platelet count, and the PLR was calculated by division of platelet and lymphocyte count. The inflammatory marker MPV/PLT, further referred to as PVPR (platelet volume to platelet ratio), was defined as the ratio of MPV (medium platelet volume) to platelet count.

### Statistical Analysis

Inflammatory markers were dichotomized based on receiver operator characteristic (ROC) analysis, with a binary state variable split at the median of survival. The best cut-off value was obtained by maximizing the Youden index (or equivalently by optimizing the sum of sensitivity and specificity). In case of nonsignificant results of ROC analysis, the median value of the analyzed marker was used as a tradeoff. Overall survival (OS) was defined as the time from the diagnosis of stage IV disease to death, and progression free survival (PFS) was calculated from the initiation of chemotherapy to either death or disease progression appointed by the leading physician. Otherwise, the subject’s case was censored. Categorical variables were reported with frequencies and percentages, and continuous variables were characterized by medians and interquartile ranges. The Kaplan–Meier method was used to estimate the medians of PFS and OS, and the comparison of survival functions between two or more independent groups was performed by applying a log-rank test. Cox regression analysis was employed to evaluate the impact of selected variables on OS and PFS. Inflammatory markers that were statistically significant in univariable models were designated for further multivariable analysis. Possible redundancy of inflammatory markers was verified based on the pairwise calculated Spearman’s rank correlation coefficients and Cohen’s kappa coefficients under criterion of redundancy detection: rho greater than 0.6 and kappa greater than 0.5, simultaneously. Selected inflammatory markers as well as other plausible clinically meaningful variables were considered as admissible for inclusion in the multivariable Cox regression analysis. The proportional hazard assumption in all multivariable models was tested with visual inspection of Schoenfeld residuals, and was formally complemented with omnibus chi-squared goodness of fit tests, relating failure time to covariate values. Additional evaluation of risk models was performed based on Harrell’s C-index (concordance index) and a value significantly greater than 0.5 was regarded as indicating that the estimated risk scores are good at determining which of the two patients will experience the endpoint first. To get a reliable and robust estimation of the contribution of inflammatory markers to the predictions of OS and PFS, and to verify the consistency of the results, a variety of survival models were fitted covering the wide spectrum of sets of covariates admissible for adjustment. ECOG, the well-known strong risk factor, yet violating the proportional hazard assumption (PH), could not be directly incorporated in multivariable models; instead, the stratification on the ECOG variable was performed. Consequently, it was not possible to obtain a hazard ratio value for the effect of ECOG in multivariable analysis, due to different baseline hazard function for categories 0, 1, and 2. However, roughly parallel log-log survival curves allowed the calculation of the size effect for ECOG in the univariable analysis. The study was exploratory, and no correction was applied for multiple statistical testing. Analyses were done in the SPSS version 27 (IBM corporation, Armonk, NY, USA) and R software (Development Core Team, Vienna, Austria, version 4.0.4). All tests were two-sided and statistical significance was defined as *p* < 0.05.

## 3. Results

### 3.1. Patient and Tumor Characteristics

Final analysis consisted of one hundred and fifty-five patients. Patient characteristics are reported in Table 1. To analyze the possible impact of administered chemotherapy regime, all patients were divided into four groups based on the intensity of chemotherapy (monotherapy, two-drug regimen, tree-drug regimen, and trastuzumab-containing regimen). Prior gastrectomy and prior perioperative treatments (e.g., neoadjuvant chemotherapy or adjuvant radio-chemotherapy) were coded as binominal covariate. Median NLR was 3.26 (95% CI 2.2–5.06) and median PVPR was 0.34 (95% CI 0.24–0.46).

### 3.2. Survival Analysis

Median OS and PFS for the entire analyzed population were 10.6 months (95% CI 9.4–11.9) and 5.5 months (95% CI 4.3–6.7), respectively. The ROC analysis summarizing the performance of the classifier was significant only for NLR and PDW, for which the best cut-off point was used as a threshold, while the inflammatory markers median value was applied in the remaining cases. The area under curve (AUC) for NLR was 0.651 (95% CI, 0.559–0.742; *p* = 0.002). Optimal cut-off ratio calculated with the Youden index for NLR was 3.99 (Youden index 0.259, sensitivity 0.519, specificity 0.740, PPV (positive predictive value) 0.678, NPV (negative predictive value) 0.593). For PVPR, median was used as cutoff (sensitivity 0.580, specificity 0.534, PPV 0.541, NPV 0.574). 

In the univariate Cox regression analysis, the dichotomized NLR, PDW, and PVPR as well as performance status were statistically significant (Figure 1). Higher NLR and lower PDW and PVPR were associated with inferior OS and PFS. 

A strong correlation between PDW and PVPR was noted, as well as a weak to minimal correlation between either NLR-PVPR and NLR-PDW (Table 2). Based on this observation, a concurrent use of PDW and PVPR was ruled out due to the high risk of redundancy. Additionally, Cohen’s κ was run to assess the agreement between dichotomized inflammatory markers, and based on the result finally PVPR together with NLR was chosen as PVPT is less correlated with NLR than PDW (Appendix A).

Multiple Cox regression models with various covariates showed a statistically significant impact of both NLR and PVPR on OS, but only NLR had an impact on PFS (Table 3). In the most adjusted model, patients with high PVPR had a 44% lower risk of death compared with individuals with a low PVPR score (HR = 0.56; 95% CI: 0.33–0.93), and the results in all models were robust showing HR oscillating from 0.53 to 0.66. The risk of death in patients with high NLR values was almost two-fold higher than in low NLR patients (HR = 1.95; 95% CI: 1.17–3.24) in Model 4 (fully adjusted). The results did not generally differ between models. To avoid violation of proportional hazard assumption, models were stratified for performance status (ECOG). Other than this, no violation of Cox regression assumptions was noted. Harrell’s C-index validating the predictive ability of a survival models confirmed that included variables prognose the time to an event sufficiently well, depicting the fraction of pairs where the observation with the higher survival time has the higher probability of survival predicted by the model at a level of at least 0.62.

To complement the results of the Cox regression analysis, Kaplan–Meier survival curves based on designated inflammatory markers were constructed (Figure 2, Figure 3, Figure 4 and Figure 5). A significant difference was found between high and low NLR and PVPR (Appendix A).

To finally validate this model, we developed a combined NLR and PVPR score (direction on influence of NLR and PVPR is opposite). The Kaplan-Meier curve with log-rank test (Figure 6) showed statistical and clinical significance.

## 4. Discussion

Our study showed that NLR and PVPR score could be used as a novel and unique predictor of survival in stage IV gastric cancer patients receiving standard cytotoxic chemotherapy. 

Lack of accurate predictive factors hinders optimization of cytotoxic chemotherapy in gastric cancer. Our study defined a promising marker, but the exploratory nature of this analysis requires prospective validation. Chemotherapy might be tailored for individual GC patients when a proven and cheap predictive factor is available. 

We have shown that both NLR and PVPR had a significant impact on OS and PFS in the first-line treatment of gastric cancer. These markers also did not seem to correlate with each other, so it should be concluded that they reflect two distinct realms of inflammatory response. Accordingly, a combined score of NLR and PVPR seems to enhance each parameter’s prognostic strength. 

Patients analyzed in the study were treated with cytotoxic chemotherapy appropriate for a given time, with a shift from a 3-drug regimen (e.g., EOX, mDCF) to a 2-drug regimen (e.g., XELOX), resulting from changes in gastric cancer treatment guidelines. In our opinion, this is a very interesting setting and further retrospective analysis should be conducted to analyze the difference in predictive factors pre and post shift. 

The NLR is a well-established marker of inflammation in multiple malignancies and other diseases [30,31,32]. There are several explanations of the detrimental impact of impaired neutrophil-lymphocyte balance in cancer patients. Activated neutrophils promote tumor growth by producing proangiogenic factors [33], such as vascular endothelial growth factor (VEGF) [34], interleukin-8 [35], and matrix metalloproteinases [36]. On the other hand, neutrophils might also suppress the anti-tumor immune mechanisms by suppressing both natural killers and CD4+ and CD8+ T cells [37]. Contrary to the negative influence of high neutrophil levels, large populations of lymphocytes, both peripheral [38] and tumor-infiltrating [39], seem to correlate with improved survival of cancer patients. In numerous studies and across various malignancies, the increased peripheral blood ratio of neutrophils to lymphocytes constantly correlated with detrimental outcomes of cancer patients [40]. 

Recent insights into the interplay between cancer cells and platelets underline its importance in promoting tumor progression, by decreasing the activity of natural killer lymphocytes [41], stimulation of deep venous thrombosis [42], and formation of neutrophil extracellular traps [43]. Therefore, in numerous studies, the elevated platelet count was shown to have a detrimental effect on OS in patients with various malignancies [44,45]. Many groups have also thoroughly evaluated several indicators of platelet-related activity, such as PLR and PDW. In our study, the most critical platelet-related factor seemed to be the PVPR. The main problem with assessing the impact of platelet volume on survival in malignancies is the heterogeneity of results. High MPV correlated with poor survival in most studies, as large platelets are considered more active [46]. Large platelets also increase the risk of thromboembolic events [47], which may further decrease survival rates. On the other hand, a recent meta-analysis on PDW, which is strongly correlated with MPV, showed opposite outcomes in GC patients [48]. Two trials showed no correlation between OS and PDW in gastric cancer in the analyzed group, and one showed even improved survival in patients with higher PDW. This phenomenon requires further analysis. 

We are aware of the limitations of our study. The analysis was retrospective, and all data were gathered manually. There was an initial bias due to only analyzing patients that received at least one cycle of chemotherapy, so the group of patients with exceptionally bad prognosis was omitted. Nevertheless, our study, comprising a real-world population of advanced GC patients, treated with standard chemotherapy, allowed for the definition of a novel, unique, and readily available prognostic factor in clinical practice.

## 5. Conclusions

In summary, the NLR and PVPR score (high NLR and low PVPR) is a strong predictor of detrimental outcomes (both OS and PFS) in advanced, chemo-naïve GC patients undergoing first-line palliative chemotherapy. Validation of this novel prognostic factor in other populations of cancer patients is warranted. 

## Figures and Tables

**Figure 1 jcm-10-03902-f001:**
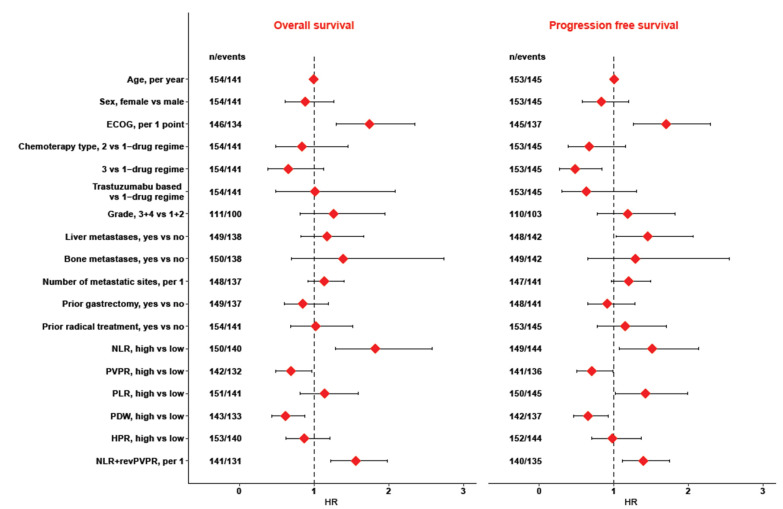
Univariable Cox regression.

**Figure 2 jcm-10-03902-f002:**
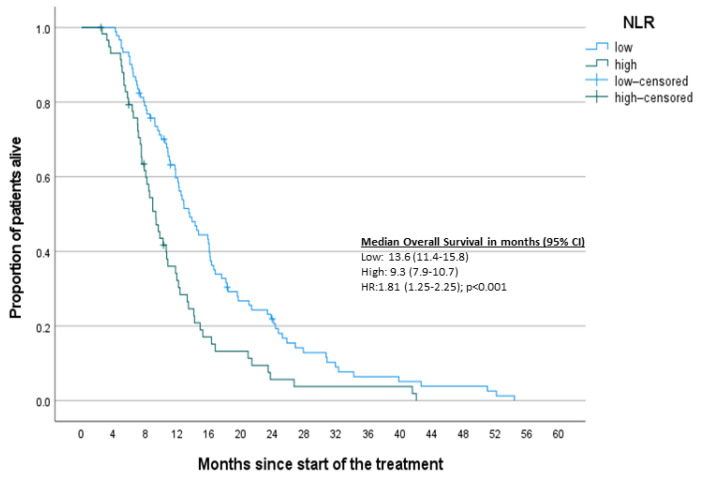
Kaplan–Meier curves for OS-NLR.

**Figure 3 jcm-10-03902-f003:**
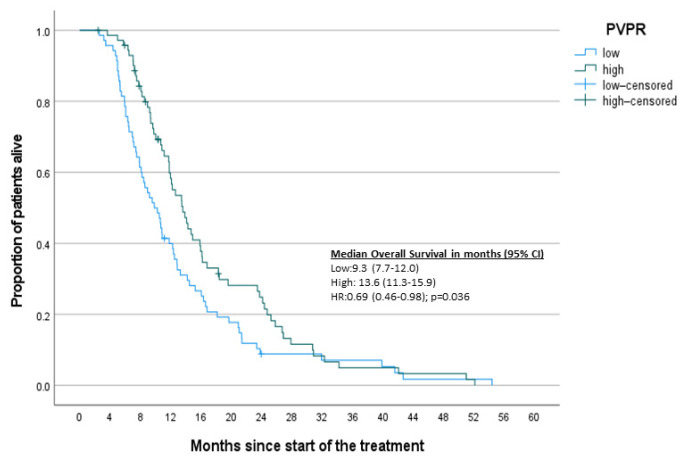
Kaplan–Meier curves for OS-PVPR.

**Figure 4 jcm-10-03902-f004:**
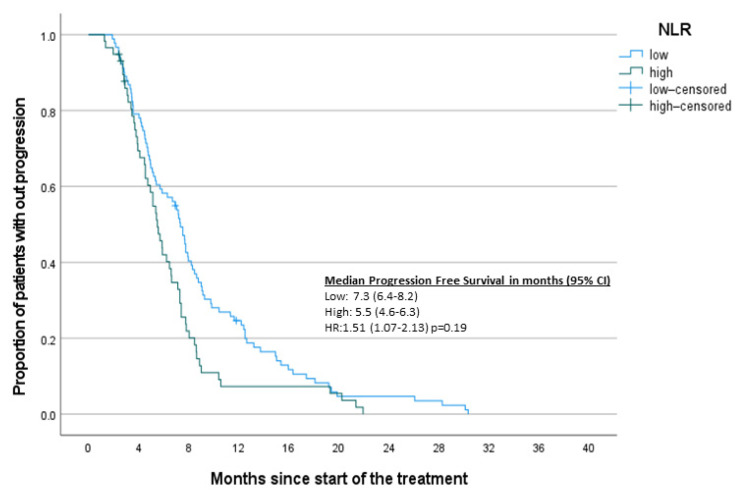
Kaplan–Meier curves for PFS-NLR.

**Figure 5 jcm-10-03902-f005:**
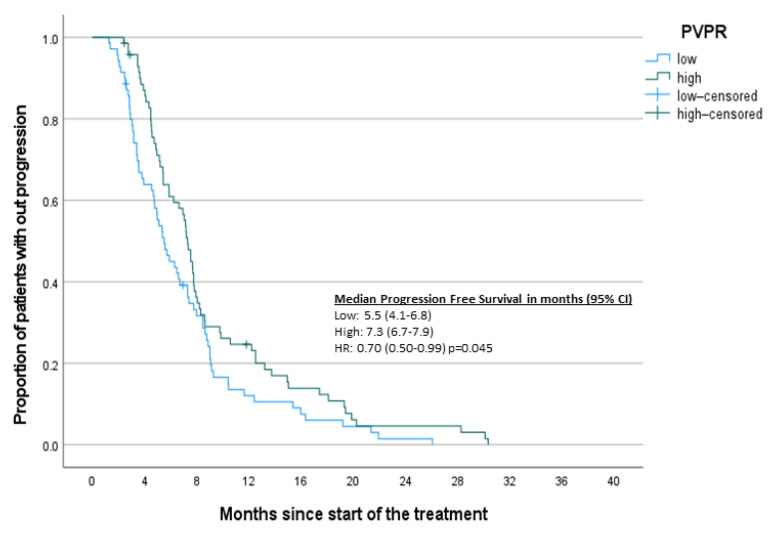
Kaplan–Meier curves PFS-PVPR.

**Figure 6 jcm-10-03902-f006:**
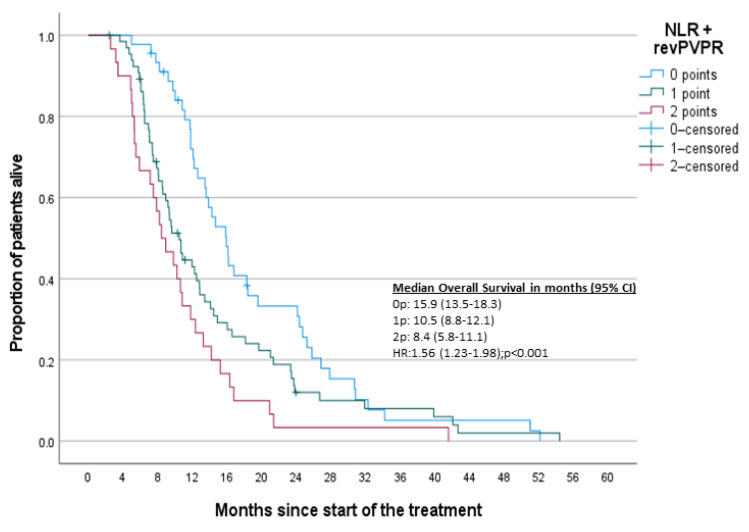
Kaplan–Meier curves for NLR and PVPR-OS. Note: PVPR is reversed due to opposite direction of influence.

**Table 1 jcm-10-03902-t001:** Characteristics of categorical and continuous variables.

**Variable**	***n***	**q2 (q1–q3)**	**min–max**
Age	155	62 (56–70)	32–82
OS	102	324 (191.25–474)	65–1602
PFS	154	158 (84.75–239)	8–880
NLR	151	3.26 (2.2–5.06)	0.79–13
PVPR	143	0.34 (0.24–0.46)	0.11–1.28
PLR	152	203.21 (139.11–279.42)	52.09–938.24
HPR	154	0.37 (0.27–0.53)	0.10–1.79
PDW	144	11.85 (10.67–13.53)	8.4–20.9
**Variable**	**Categories**	***n***	**%**
Gender	male/female	109/46	70.3/29.7
ECOG †	0/1/2/NR	24/92/31/8	15.5/59.4/20/5.2
Grade	1/2/3/4/NR	3/30/78/1/43	1.9/19.4/50.3/0.6/27.7
HER2 expression	0-2/3/NR	85/16/54	54.8/15.8/34/8
Prior radical treatment ‡	no/yes/NR	41/114/3	26.5/73.5/1.9
Prior gastrectomy	0/1/NR	87/63/5	56.1/40.6/3.2
Bone metastases	no/yes/NR	141/10/4	91/6.5/2.6
Liver metastases	no/yes/NR	98/52/5	63.2/33.5/3.2
Number of metastatic sites	0/1/2/3/NR	21/91/27/10/6	13.5/58.7/17.4/6.5/3.9
Type of chemotherapy Δ	1/2/3/4	19/59/62/15	12.3/38.1/40/9.7

Δ—1-one-drug regime, 2-two-drug regime, 3-three-drug regime, 4-trastuzumab-based regime. ‡ chemotherapy used in radical setting prior to dissemination (e.g., Macdonald regime). †—performance status (Eastern Cooperative Oncology Group). NR-not reported. NLR—neutrophil to lymphocyte ratio, PVPR—medium platelet volume to platelets ratio, PLR—platelets to lymphocyte ratio, HPR—hemoglobin to platelets ratio, PDW—platelets distribution width.

**Table 2 jcm-10-03902-t002:** Sperman rank-order correlation of inflammatory markers.

	1	2	3	4	5
1. NLR	1				
2. PVPR	−0.17 *	1			
3. PDW	−0.24 **	0.67 **	1		
4. HPR	−0.14	0.93 **	0.54 **	1	
5. PLR	0.40 **	−0.28 **	−0.17	−0.34 **	1

Note: Analysis of multicollinearity between inflammatory marker. Results closer to zero have weaker correlation * *p* < 0.05 (bilateral), ** *p* < 0.01 (bilateral), NLR—neutrophil to lymphocyte ratio, PVPR—medium platelet volume to platelets ratio, PLR—platelets to lymphocyte ratio, HPR—hemoglobin to platelets ratio, PDW—platelets distribution width.

**Table 3 jcm-10-03902-t003:** Multivariable Cox regression analysis.

	Model 1 *^a^	Model 2 *^b^	Model 3 *^c^	Model 4 *^d^
OS				
*n*/Events ‡	131/122	133/124	126/117	93/85
Harrell’s C-index (95% CI) ±	0.65 (0.59; 0.71)	0.67 (0.61; 0.73)	0.64 (0.58; 0.7)	0.65 (0.58; 0.73)
p_PHA_ ¤	0.269	0.47	0.517	0.274
PVPR	0.66 (0.45–0.98)	0.66 (0.44–0.98)	0.63 (0.41–0.96)	0.56 (0.33–0.93)
NLR	1.74 (1.18–2.57)	1.78 (1.2–2.64)	1.85 (1.23–2.78)	2.02 (1.2–3.37)
PFS				
*n*/Events ‡	130/125	132/127	125/120	92/87
Harrell’s C-index (95% CI) ±	0.62 (0.56; 0.68)	0.65 (0.59; 0.71)	0.65 (0.6; 0.71)	0.64 (0.57; 0.72)
p_PHA_ ¤	0.64	0.79	0.587	0.395
PVPR	0.73 (0.5–1.07)	0.71 (0.48–1.05)	0.71 (0.47–1.06)	0.8 (0.49–1.33)
NLR	1.5 (1.01–2.22)	1.68 (1.13–2.49)	1.98 (1.3–3.04)	2.02 (1.19–3.45)

Note: Multiple Cox regression models showed consistent impact of NLR and PVPR on OS. * stratified by ECOG and additionally adjusted to: ^a^ Model 1: age, gender, liver metastases, bone metastases; ^b^ Model 2: age, gender, prior radical treatment, chemotherapy type; ^c^ Model 3: liver metastases, bone metastases, chemotherapy type, prior gastrectomy, sum of metastases; ^d^ Model 4: age, gender, liver metastases, bone metastases, prior radical treatment, chemotherapy type, prior gastrectomy, sum of metastases. ¤ *p* value from overall omnibus test verifying proportional hazard assumption; ‡ number of cases/ number of non-censored events; ± Harell’s concordance index to assess goodness of fit of various models. NLR—neutrophile to lymphocyte ratio. PVPR—median platelet volume to platelets ratio.

## Data Availability

The data presented in this study are available on request from the corresponding author. The data are not publicly available due to privacy concern.

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
