# Peer review of "Combined Neutrophil-to-Lymphocyte and Platelet-Volume-to-Platelet Ratio (NLR and PVPR Score) Represents a Novel Prognostic Factor in Advanced Gastric Cancer Patients"

_jcm, 2021, doi:10.3390/jcm10173902_

Round 1

Reviewer 1 Report

Comments to the Author

General comments
In this paper, Konopka et al propose a novel prognostic tool for survival rate in Gastric Cancer.

This is a retrospective study with a deep statistical analysis that show that changes in NLR and PVPR among other markers can predict OS.

The topic should be of interest, but the conclusion that these parameters can be used in clinical practice seems to be too much. I suggest to make a discussion less speculative.

Specific comments

FIGURES and TABLES:

Please provide more detailed figure and table legends with the description of data presented.

Author Response

Hello,

Thank you very much for your review.

I am a bit unexperienced and I am not sure if I should respond you as soon as possible and send the revised manuscript later or the opposite.

  • I have received conflicting comments from you and your colleague regarding conclusions of the article. Main concern in the second review is that I did not show enough clinically meaningful comments. I believe that both opinions are valid and represent two distinct categories of possible readers. On account of that, I have to rewrite the discussion and put emphasis that, due to the nature of this study, it is a hypothesis generating trial and the described inflammatory marker might be useful in future. I am quite proud of my work here, but now I see that my enthusiasm might made me overconfident and some of my conclusions need refining. I do believe that there is still a place in modern oncology for 3-drug regimes, such as mDCF, and a better understanding of the predictive factors might be a tool to distinguish a patient that requires more intensive treatment.
  • Figures and tables require a more precise legend and I will rewrite those.

Best regards 

Reviewer 2 Report

In this retrospective study, Konopka et al. aimed to comparatively evaluate the utility of multiple peri-treatment laboratory data indices to predict postoperative long-term outcomes in patients undergoing chemotherapy for advanced-stage gastric cancer (GC). They concluded that combination of neutrophil-to-lymphocyte (NLR) and platelets-volume-to-platelet ratio (PVPR) can serve as a valuable prognosticator for disseminated GC.

Despite being nicely organized, this article has some un-ignorable concerns.

Major points:

This is one of several similar studies assessing the usefulness of any of a number of laboratory indices in predicting survival outcomes in patients undergoing treatment for solid tumors. Although the authors successfully demonstrated that the pretreatment indices were able to predict post-treatment prognosis in patients with advanced GC, they provided few concrete guidance as to how these values facilitate selecting the optimal therapeutic strategy for these patients. I suppose that this is truly the crux of the issue with studies such as these - how are these indices clinically useful to the physicians or surgeons faced with making treatment decisions for patients receiving systemic chemotherapy for GC?

Minor points:

  • There were too many missing data (Table 1).
  • The study is limited by small sample size (n = 155).
  • Redundant descriptions, inconsistent formats and poor English can solely be the reason for rejection.

Author Response

Thank you very much for your review.

I am a bit unexperienced and I am not sure if I should respond you as soon as possible and send the revised manuscript later or the opposite.

  • I have received conflicting comments from you and your colleague regarding conclusions of the article. Main concern in the second review is that my final thoughts are too speculative and I should tone it down. I believe that both opinions are valid and represent two distinct categories of possible readers. On account of that, I have to rewrite the discussion and put emphasis that, due to the nature of this study, it is a hypothesis generating trial and the described inflammatory marker might be useful in future. I do believe that there is still a place in modern oncology for 3-drug regimes, such as mDCF, and a better understanding of the predictive factors might be a tool to distinguish a patient that requires more intensive treatment. My trial, due to it size and nature, is not meaningful enough to make such a guideline, but I think that this hypothesis is a good starting point for more targeted trials.
  • Missing data: Most of the missing data comes from two sources. In the past, the diagnosis of disseminated malignancies in Poland was not done properly and in many situations only fine needle biopsy was performed – due that histopathological reports are missing many of the important data, such as grade of disease. Second source is the problem with Her2 positive gastric cancer. Due to very unwise reimbursement rules in Poland, trastuzumab is only available for patients in very good general condition, so that there was a tendency not to check Her2 when trastuzumab would not be available. It changed nowadays, but sadly I am not able to rerun this test.
  • Sample size: No doubt here, but I have gathered all date I was able to and this group was sufficient to adequately prove my hypothesis. Further increasing population here is futile due to the type of study: retrospective analysis with out correction for multiple statistical tests. I certainly hope that I will be able to verify this finding in a second registered report on a different population of more recent patients.
  • Redundancy : I was afraid that readers might be confused with the statistical methods I used to deliver my results, but I will improve it.
  • English: I did use professional proofreading, but the manuscript after revision will be checked again.
  • Inconsistent formats require my improvement.

Best regards

Round 2

Reviewer 2 Report

Although the study concept is unchanged, the authors have made an effort to improve soundness of the draft. Now it is acceptable.